# Mapping dementia research in Indonesia: A scoping review of evidence, gaps, and future directions

Anna Tjin[1,2*], Fasihah Irfani Fitri[3,4], Michael Maitimoe[5], Diany Syafitri[6,7], Shofia Mawaddah[7,8], Tung Le[9], Firda Aminy Ma'ruf[10], Sarah Bauermeister[11], Robert Stewart[1,12], Iracema Leroi[4,13], Asri Maharani[14]

1 Psychological Medicine, Institute of Psychiatry, Psychology & Neuroscience, King's College London, London, United Kingdom, 2 Department of Japanese Studies, Faculty of Arts and Social Sciences, National University of Singapore, Singapore, 3 Department of Neurology, Faculty of Medicine, Universitas Sumatera Utara, Kota Medan, Sumatera Utara, Indonesia, 4 Global Brain Health Institute, Trinity College Dublin, Dublin, Ireland, 5 Alzheimer's Indonesia, South Jakarta City, Jakarta, Indonesia, 6 Sultan Agung Islamic University Indonesia, Semarang, Central Java, Indonesia, 7 Department of Psychology, Institute of Psychiatry, Psychology & Neuroscience, King's College London, London, England, 8 Universitas Negeri Medan, Medan, North Sumatra, Indonesia, 9 Mental Health Research Group, Division of Nursing, Midwifery and Social Work, School of Health Sciences, University of Manchester, Manchester, United Kingdom, 10 Barts Cancer Institute, Queen Mary University of London, London, United Kingdom, 11 Department of Psychiatry, University of Oxford, Oxford, United Kingdom, 12 South London and Maudsley NHS Foundation Trust, London, United Kingdom, 13 School of Medicine, Trinity College Dublin, Dublin, Ireland, 14 Manchester Academic Health Science Centre (MAHSC), Manchester, United Kingdom

* anna.tjin@kcl.ac.uk

## Abstract

Dementia, a syndrome that progressively impairs cognitive functions, is a growing global health challenge. Indonesia, the world's fourth most populous country, faces a growing dementia burden compounded by stigma and inequities in prevention, diagnosis, and care. This scoping review aims to synthesise dementia research in Indonesia, identify gaps, and propose directions for future research. Following the Joanna Briggs Institute and PRISMA-ScR guidelines, a comprehensive search across eight databases (MEDLINE, Embase, CINAHL, Global Health, Web of Science, PubMed, PsycINFO, and GARUDA) were conducted to identify studies in English and Indonesian. 105 studies were included, with most (93.3%) studies published after 2016, aligning with Indonesia's National Dementia Plan and the WHO Global Action Plan on Dementia. Studies were predominantly cross-sectional (73.3%) and concentrated in urban areas. When mapped against the WHO Global Action Plan on Dementia (2017–2025), studies clustered around seven key themes. Key risk factors examined included older age, low education, female sex, low socioeconomic status, smoking, hypertension, diabetes, and physical inactivity. Work on diagnosis, treatment, and care has expanded, particularly through validation of cognitive screening tools (e.g., MoCA-INA, BCSB-INA) and emerging use of neuroimaging and biomarkers, though implementation remains limited by cost and workforce capacity. Intervention studies are typically small-scale, short-term,

**Data availability statement:** Yes - all data are fully available without restriction; As this study is a scoping review, all data underlying the findings are derived from publicly available sources and are contained within the manuscript and its Supporting Information files.

**Funding:** The author(s) received no specific funding for this work.

**Competing interests:** The authors have declared that no competing interests exist.

and lack longitudinal evaluation. Findings consistently showed high psychosocial and financial burden, especially among female family carers, with unmet needs for training and emotional support. Finally, policy and systems-level research highlighted limited integration of dementia into primary healthcare, inadequate data infrastructure, and minimal progress in translating the 2016 National Dementia Plan into sustainable support systems. Dementia research in Indonesia has expanded, yet geographical and methodological gaps persist. Future priorities should include nationally representative studies, implementation research, and multisectoral collaborations to advance the WHO's vision of dementia as a public health priority and strengthen preparedness for its ageing population.

## Introduction

Dementia, a syndrome that progressively impairs cognitive functions (e.g., memory, judgment) [1], presents a substantial challenge to global health outcomes. In 2019, an estimated 55.2 million people worldwide had dementia. It is projected to reach 139 million by 2050 [2]. Low- and middle-income countries (LMICs) are disproportionately affected by dementia, with over 60% of People with Dementia (PwD) living in LMICs with limited service provision and health education and symptoms often seen as normal ageing [3].

Indonesia, a diverse LMIC with over 600 ethnic groups and the fourth most populous country globally [4], had 25.7 million people aged 60+ (9.6%) in 20245, with 74 million (25%) projected by 2050 [5]. A projection of global age-specific prevalences to Indonesia's ageing population estimated around 1 million PwD in 2024, with this number expected to triple over the next three decades [2]. However, this estimate was derived by applying global age-specific prevalence rates to Indonesia's ageing (60+) population [2]. The actual prevalence of dementia is likely much higher due to a lack of clear diagnosis pathways and severe stigma, which discourages diagnosis and treatment [6], while the lack of a national registry and reliance on non-representative hospital data leave existing estimates fragmented and inconsistent. Despite the growing burden of dementia, Indonesia continues to grapple with a more urgent backlog of infectious diseases, malnutrition, and maternal mortality [7], leading to the under-prioritisation of non-communicable diseases like dementia. Furthermore, Indonesia invests only 3% of its GDP in healthcare [8], a figure lower than that of other Global South countries [8], leading to critical shortages of healthcare personnel, underdeveloped health services, and an over-reliance on tertiary care [9,10]. The Indonesian Ministry of Health has developed dementia policies, including the 2016 National Dementia Plan [11,12], 2019 regulation of essential services (e.g., cognitive screening) [13], and the global action plan on the public health response to dementia [14]. However, the dementia care pathway in Indonesia is still in its early phase, with reports of gaps and inequality in dementia prevention, diagnosis, and care systems [15]. In Indonesia's limited dementia care system and communal culture, family carers often shoulder

the majority of care responsibilities [15,16]. Despite the benefits of caregiving, multiple studies reported significant psychosocial and financial consequences [17,18], emphasising the need for a more inclusive and sustainable dementia care system that supports family carers.

A key step in strengthening dementia care capacity is evaluating existing research to identify gaps and unmet needs of Pwd and carers. To date, only two reviews have examined dementia in Indonesia [19,20]. One prior review focused exclusively on dementia prevalence, while another examined family caregiving experiences, together offering valuable but fragmented insights that overlook other critical aspects such as diagnostics, interventions, and policy. This review aims to provide a comprehensive synthesis of dementia research in Indonesia, mapping its main directions in alignment with the World Health Organisation Global Action Plan on the Public Health Response to Dementia (2017–2025) [21]. It examines evidence across seven key domains: dementia as a public health priority; awareness and friendliness; risk reduction; diagnosis; treatment, care, and support; support for carers; and research and innovation, to identify existing evidence, gaps, and priorities for strengthening Indonesia's national dementia response. By highlighting study designs, geographical coverage, and thematic emphases, the review identifies knowledge gaps and proposes pathways for future research. Findings will strengthen the evidence base for a less-studied population and inform the development of a more inclusive, contextually relevant health system.

## Materials and methods

### Search strategy

This scoping review followed the Joanna Briggs Institute guide [22] and PRISMA-ScR (Preferred Reporting Items for Systematic Reviews and Meta-Analyses extension for Scoping Reviews) guidelines (check list is reported in S1 File **PRISMA Checklist**) to ensure transparent reporting of the review process [23]. The review is registered in the Open Science Framework [24].

The search was conducted across eight electronic databases (MEDLINE, Embase, CINAHL, Global Health, Web of Science, PubMed, PsycINFO, and the *Garba Rujukan Digital* (GARUDA) portal from Indonesia's Ministry of Research) from June 2024, and re-run in August 2025, without restrictions on publication year or study methods. Google Scholar was searched, and the first 40 pages were reviewed to capture regional publications and studies from local or non-indexed journals not covered by traditional databases, ensuring a comprehensive evidence base. In addition, other grey literature was screened, including government, charities, and international organisation reports, policy documents, theses, conference proceedings, preprints, and technical reports. The search was limited to studies in English and Indonesian conducted in Indonesia or that included Indonesia in multisite studies. Forward and backward citation searching was conducted for the articles included and the reference lists of literature reviews. The search strategy for EMBASE (**Table 1**) was revised for each database utilised.

### Screening, article selection, and data extraction

The review team developed and iteratively refined the inclusion/exclusion criteria (**Table 1**). Once the selection criteria were outlined, three pairs of authors conducted data searching and title/abstract screening, followed by team meetings to discuss decisions and ensure consensus. After screening, each pair reviewed full-text articles. Regular team meetings were held to address challenges and resolve disagreements. To categorise diagnoses, we followed international standards (DSM-5 and ICD-11), which classify neurocognitive disorders by primary aetiology (e.g., Alzheimer's disease, vascular dementia, frontotemporal dementia, Parkinsonism-related dementias), pre-dementia states (e.g., mild cognitive impairment), and secondary causes (e.g., stroke, traumatic brain injury). This framework has also been applied in global burden estimates and LMIC dementia research [2]. Accordingly, we grouped the Indonesian studies into three categories: (i) primary dementias, (ii) pre-dementia/cognitive impairment states, and (iii) secondary/comorbidity-related cognitive impairment.

**Table 1. Inclusion and exclusion criteria, and search terms.**

| Inclusion criteria | Exclusion criteria |
|---|---|
| Participants included adults (18+) with dementia, Alzheimer's disease, cognitive impairment or who were caring for people with the diagnosis. | Studies investigating unrelated medical conditions. |
| Studies carried out in Indonesia or partly in Indonesia as a component of a multisite study. | Studies carried out in countries other than Indonesia, not including Indonesia in multinational studies, or only include Indonesians abroad as a classification of ethnicity. |
| Original research study (quantitative, qualitative, mixed method), book chapters, grey literature (thesis/dissertations, research and committee reports, government reports), policy briefs. | Opinion pieces, book reviews, letters, editorials, or perspective pieces. |
| Studies written in English or Indonesian with full text available. | Studies not in English or Indonesian and/or with no full text available. |
| Human studies: including human participants directly (e.g., clinical trial, survey) or indirectly (e.g., human biological sample, datasets). | Non-human or animal studies (e.g., preclinical testing disease pathology, drug development) |
| **Searches** | |
| 1 | multi-Infarct/ or exp AIDS dementia complex/ or exp cognition disorders/ exp mild cognitive impairment/or (dementia or neurocognitive disorder or cognitive impairment or cognitive decline or cognition or alzheimer* disease or vascular dementia or (frontotemporal adj2 (dementia or degeneration)) or pick* disease or (lewy bod* adj2 dementia) |
| 2 | "lewy bod*" OR "DLB" OR "LBD" OR "Frontotemporal dementia" OR "FTD" OR "vascular dementia*" OR "dementia* vascular" OR "synucleinopathies" OR "cognitive impairment" |
| 3 | indonesia or "indonesia" |
| 4 | 1 AND 2 AND 3 |

## Data synthesis and analytical framework

The extracted data were charted and synthesised thematically. To ensure alignment with international priorities, the analysis was guided by the WHO Global Action Plan on the Public Health Response to Dementia 2017–2025 [21], which outlines seven strategic action areas:

(1) Dementia as a public health priority; (2) Dementia awareness and friendliness; (3) Dementia risk reduction; (4) Dementia diagnosis; (5) Dementia treatment, care and support; (6) Support for dementia carers; and (7) Dementia research and innovation. These action areas were adapted to the Indonesian research context to organise the synthesis and highlight evidence coverage, thematic emphasis, and gaps across the dementia care continuum. Although the WHO Global Action Plan on Dementia (2017–2025) guided the analytical framework, the search was not restricted by publication year in order to capture baseline evidence preceding national and global policy initiatives.

## Result

The initial literature screening (**Fig 1**) yielded 4,494 articles. 1762 articles received title and abstract screening, and 105 met the inclusion criteria and were included in this review.

## Study characteristics

A detailed table of the included studies, including author, year, study category, dementia type, study type, location, sample size, and settings, is provided in S1 Table. A summary of study characteristics is shown in **Table 2**.

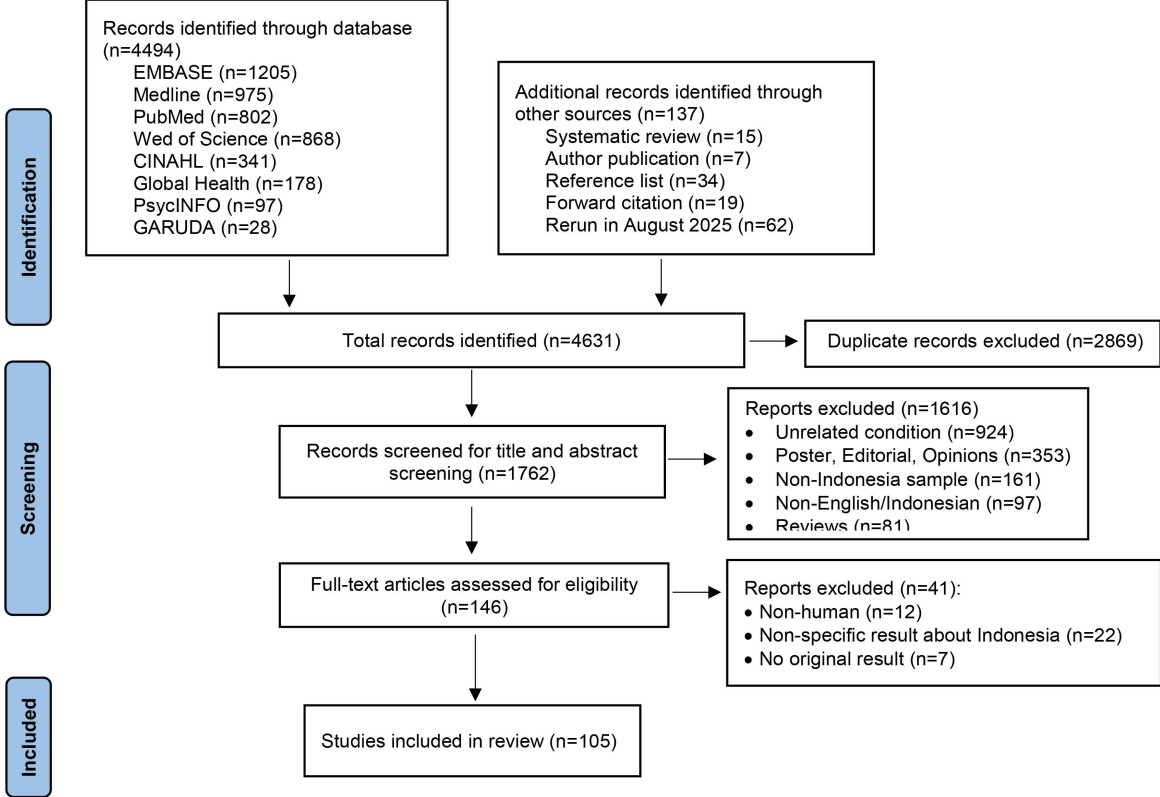

**Fig 1. Flowchart.**

As summarised in Table 2, only seven of the 105 studies were published before 2016, while the majority were produced during the period aligned with Indonesia's National Dementia Plan and the WHO Global Action Plan on Dementia (2017–2025). The sample sizes ranged from 4 to 6,755 participants. Most studies (n = 52) (**Fig 2**) were conducted in major cities, such as the capital city of Jakarta, special region of Yogyakarta, and Surabaya, and, on Java Island. Eleven were country-wide, thirteen had multiple sites in Indonesia, and ten were multinational studies. Indonesia, an archipelagic nation of over 17,000 islands and home to more than 270 million people, exhibits considerable geographic and socio-economic diversity [25]. Approximately 70% of the population resides on Java, which, despite comprising only about 7% of the country's land area, is the most urbanised and economically developed region [26]. In contrast, eastern provinces such as Papua, Maluku, and Nusa Tenggara remain less densely populated and face significant disparities in healthcare access, infrastructure, and representation in research [26]. This uneven distribution likely contributes to the concentration of dementia research in Java and the relative paucity of studies from rural and remote regions.

The majority of studies (n = 53) focused on general or unspecified dementia, while nine specifically examined Alzheimer's disease (AD), three investigated Vascular Dementia (VD), two explored Parkinsonism-related dementia, and one focused on Frontotemporal Dementia (FTD). Several studies also addressed cognitive impairment in other contexts, including mild cognitive impairment (MCI) (n = 9), traumatic brain injury (TBI) (n = 1), and stroke (n = 2). Study designs were diverse, comprising policy analysis (n = 1), randomised controlled trials (RCTs) (n = 2), case reports (n = 2), mixed-method studies (n = 2), case-control (n = 5), qualitative studies (n = 5), quasi-experimental (n = 5), and cohort studies (n = 6). The most common design was a quantitative cross-sectional study (n = 77), with probability sampling most frequently used (n = 85). Most research focused on prevalence and risk factors (n = 38), followed by studies on diagnostic tool development

Table 2. Study characteristics.

| Year | n | Study design | n |
|---|---|---|---|
| 1982-2000 | 2 | Policy analysis | 1 |
| 2001-2015 | 5 | Randomised control trial | 2 |
| 2016-2020 | 19 | Case report | 2 |
| 2021-2025 | 79 | Mixed method | 2 |
| *Study location* | | Case-control | 5 |
| Multisite (Indonesia) | 13 | Qualitative | 5 |
| National | 11 | Quasi-experimental | 5 |
| Multinational | 10 | Cohort | 6 |
| Java | 22 | Cross-sectional | 77 |
| Jakarta | 20 | *Sampling methodology* | |
| Yogyakarta | 10 | Non-probability sampling | 20 |
| Sumatra | 6 | Probability sampling | 85 |
| Sulawesi | 5 | **Study types** | |
| Bali | 4 | Theme 1: Public Health Priority | 1 |
| Papua (West New Guinea) | 2 | Theme 2: Awareness and Friendliness | 10 |
| West Nusa Tenggara | 2 | Theme 3: Risk Reduction | 38 |
| *Diagnosis* | | Theme 4: Diagnosis | 16 |
| Pre-dementia/ cognitive decline | | Theme 5: Treatment, Care, and Support | 17 |
| Mild cognitive impairment (MCI) | 9 | Theme 6: Support for Carers | 12 |
| Cognitive impairment (non-specified) | 25 | Theme 7: Research and Innovation | 11 |
| Broad dementia categories | | **Study setting** | |
| Frontotemporal dementia (FTD/FTLD) | 1 | Non-governmental organisation | 1 |
| Parkinsonism-related dementias | 2 | Care homes/ nursing centres | 6 |
| Vascular dementia (VD) | 3 | Population-level | 7 |
| Alzheimer's disease (AD) | 9 | University-based/ research facilities | 17 |
| Dementia (unspecified/ general) | 53 | Hospital/clinical-based | 32 |
| Secondary/ comorbidity-related cognitive impairment | | Community | 42 |
| Stroke-related cognitive impairment | 2 | | |
| Traumatic brain injury (TBI/mTBI) | 1 | | |

(n = 17) and those investigating interventions and management strategies (n = 17). Several studies explored carers' experiences (n = 11) and the pathophysiology and biomarkers of dementia (n = 11), while a smaller number addressed awareness and attitudes (n = 10) or evaluated health policy (n = 1).

Fig 3 presents the distribution of dementia research in Indonesia by theme and study design following the WHO Global Action Plan on the Public Health Response to Dementia 2017–2025 [21]. Notably, studies mapped to the WHO Global Action Plan domains were mostly (93.3%) published after 2016, reflecting the emergence of policy-aligned research priorities in the Indonesian context. The research landscape is dominated by *Theme 3*: *Risk Reduction*, followed by moderate activity in *Theme 5: Treatment, Care, and Support* and *Theme 4: Dementia Diagnosis*. Activity is lower for *Theme 2: Dementia Awareness and Friendliness* and *Theme 6: Support for Dementia Carers*, and lowest for Theme 7: Dementia Research and Innovation. Research remains notably scarce for *Theme 1: Dementia as a Public Health Priority*, indicating a major gap in policy and system-level studies. From a methodological perspective, cross-sectional designs overwhelmingly dominate across all themes, reflecting a strong preference for descriptive and associational research over

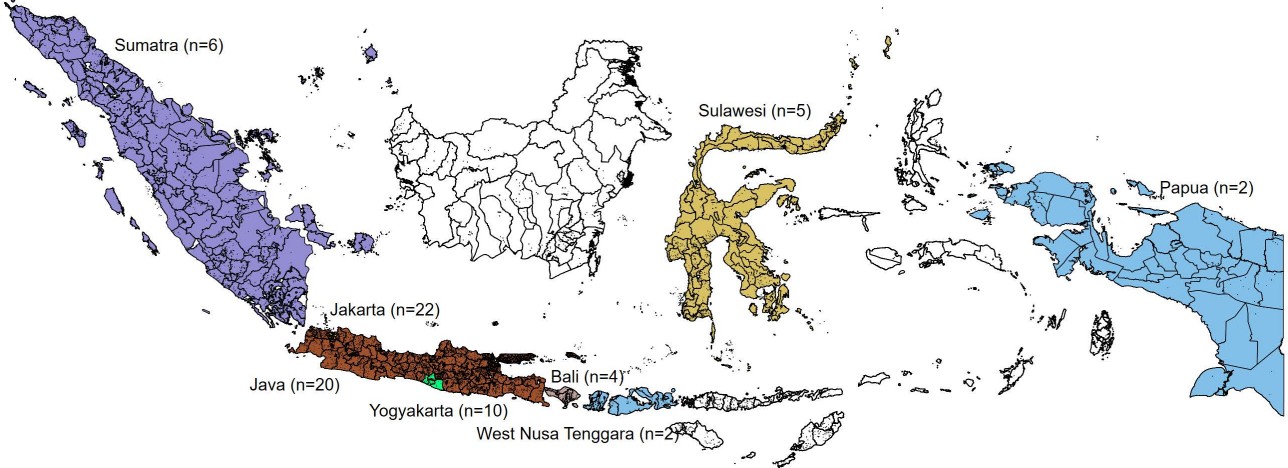

**Fig 2. Geographic distribution of dementia research in Indonesia.** Map redrawn by the authors using openly licenced geographic boundary from openfreemap.org; the figure is distributed under the Creative Commons Attribution (CC BY4.0) licence.

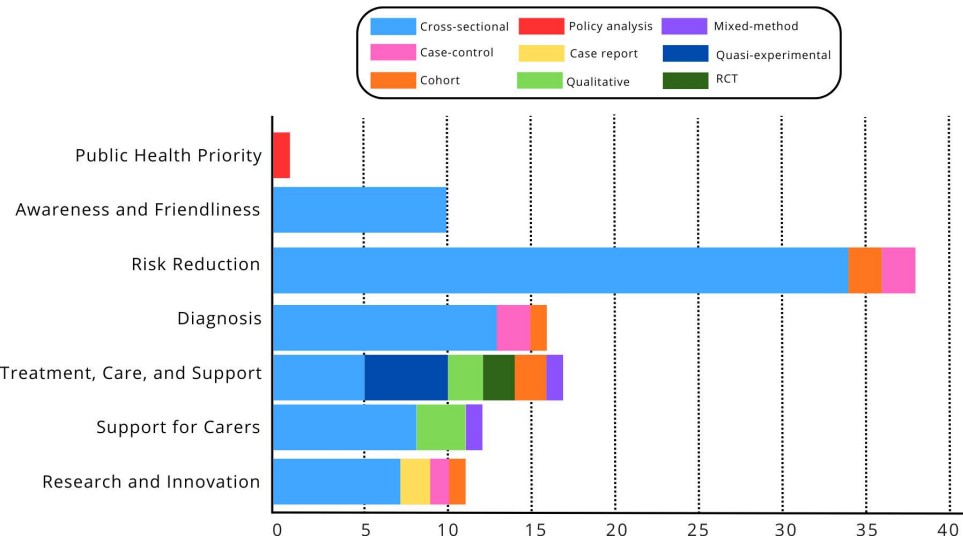

**Fig 3. Distribution of dementia research in Indonesia by theme and research characteristics.**

longitudinal or interventional approaches. Other designs, i.e. RCTs, qualitative studies, case reports, and cohort studies, appear far less frequently, revealing limited methodological diversity. This pattern suggests that most dementia research in Indonesia remains exploratory and descriptive, with few studies addressing causal mechanisms or the effectiveness of interventions. A strategic shift toward more longitudinal, experimental, and mixed-method research is essential to strengthen the evidence base for dementia policy, care, and prevention.

**Theme 1: Dementia as a public health priority.** The Alzheimer's Association Policy Brief [27], the only publication addressing the prioritisation of dementia as a public health concern in Indonesia, underscores the escalating challenges associated with the country's ageing population, with projections estimating that approximately four million people will be

living with dementia by 2050. The annual cost of care per person, ranging from IDR 12–120 million (≈USD 723–7,237), substantially exceeds the national income per capita of IDR 59 million (≈USD 3,558). Despite the introduction of the 2016 National Dementia Plan [11], substantial gaps persist in the standardisation of care, diagnosis, management, and clinical guidelines. The Association advocates for stronger coordination within referral systems, harmonised screening tools, and enhanced collaboration between government bodies, the private sector, and civil society to strengthen dementia care and improve the quality of life (QoL) of people with dementia and their carers.

Although limited research has positioned dementia as a public health priority, several studies have proposed strategic approaches to mitigate dementia risk. Public health initiatives have centred on smoking cessation, hypertension control, hearing preservation, and mental health support. Azwar and Setiati [28] call for nationwide health promotion campaigns and accessible screening services, particularly within underserved regions. Breuer et al. [29] highlight the need to address inequities in diagnosis and treatment, particularly in rural and low-income communities, through subsidised diagnostic services, mobile memory clinics, and carer training programmes. Furthermore, Dawes et al. [30] recommend the integration of hearing and vision assessments within dementia care pathways, while Farina et al. [31] propose embedding dementia education into school curricula to raise awareness, reduce stigma, and cultivate a dementia-inclusive society.

**Theme 2: Dementia awareness and friendliness.** Ten studies [30–39] examined dementia awareness and friendliness, all employing cross-sectional designs, including one multinational, one multisite, two national, and six local studies conducted primarily in Jakarta and Yogyakarta. Across these studies, participants demonstrated varying levels of understanding of dementia, with misconceptions and inconsistent terminology commonly reported. One national survey found that 86.3% of respondents were unfamiliar with the terms "*dementia*" or "*Alzheimer's disease,*" instead referring to the condition as *pikun* (forgetfulness), with awareness levels higher among younger and more educated participants in urban areas [31]. Another study involving 354 families in Jakarta found that forgetfulness was identified as the most concerning symptom due to its potential to interfere with daily activities [32].

Among healthcare professionals, knowledge and confidence levels varied. In studies of physiotherapists and community health workers [30,33–37], 15–37% correctly identified key dementia symptoms, more than half reported no prior training, and only 36% had experience treating patients with dementia [25]. Training gaps were also evident in long-term care facilities, where only 14% of staff had received sensory care training. Studies involving nurses reported generally positive attitudes toward people with dementia, and dementia education programmes for community health workers were associated with significant improvements in knowledge (p = 0.002) [34].

Recent research has also focused on developing and validating dementia-related instruments. The Indonesian version of the Confidence in Dementia (CODE) scale showed excellent content validity (S-CVI = 1.00), moderate construct validity (r = 0.526–0.633), and good internal consistency (Cronbach's α = 0.770) [38]. The Indonesian Approaches to Dementia Questionnaire (ADQ) demonstrated moderate to high subscale reliability (Hope α = 0.552; Personhood α = 0.701) and moderate item–total correlations (0.261–0.588), although several items were invalid, indicating a need for further refinement and contextual adaptation [39].

**Theme 3: Dementia risk reduction.** A total of 38 studies investigated dementia prevalence and risk factors in Indonesia [6,28,40–75]. Most were cross-sectional (n = 34), while two cohort [40,41] and two case–control studies [28,42] were identified. Reported prevalence varied widely, from 2.7% [41] to 27.9% [6], reflecting methodological and diagnostic heterogeneity across regions. Only three studies used nationally representative data from the Indonesia Family Life Survey (IFLS) [41,43,44], estimating that 1.3 million Indonesians were living with dementia in 2015, with a 2.7% prevalence disproportionately affecting women (60.7%). However, the IFLS excludes parts of eastern Indonesia, limiting generalisability. Regional studies from Java and Sumatra reported substantially higher prevalence: 27.9% using the 10/66 diagnostic schedule in Jakarta and North Sumatra [6], 29.2% in West Java [45], and 91.8% cognitive impairment among adults aged 60+ in Jakarta using the Clock Drawing Test [46]. In Banda Aceh, a community-based survey using the Saint

Louis University Mental Status (SLUMS) examination found 48.8% of older adults had mild neurocognitive disorder and 21% had dementia [66], highlighting regional variation in cognitive impairment prevalence across Indonesia.

Across studies, consistent risk factors included older age, low education, female sex, low socioeconomic status, smoking, hypertension, diabetes, stroke, depression, anxiety, physical inactivity, and low social engagement [28,42,46–50,62–65,67–71,74,75]. Consistent with these findings, a large pooled analysis including Indonesia reported that subjective cognitive decline affected approximately one quarter of adults aged ≥60, with substantially higher prevalence among individuals with lower education, populations in low- and middle-income countries, and marked heterogeneity across studies [72]. Fitri et al. (2024) [51] found vascular dementia as the predominant subtype (62.1%), followed by Alzheimer's disease (25.3%). In the same study, head trauma, hearing loss, and chronic obstructive airway disease significantly increased the risk [48]. A case–control study in Jakarta identified smoking, hypertension, diabetes, hearing loss, and depression as modifiable risk factors [28]. Another study linked vitamin D deficiency in patients with type 2 diabetes to mild cognitive impairment [42]. Conversely, higher BMI was paradoxically associated with a 40% lower risk of cognitive impairment, possibly reflecting reverse causality due to weight loss in preclinical dementia [50,63].

Several studies examined the role of frailty, psychosocial, and lifestyle factors. Frailty showed a moderate negative correlation with cognition ($\rho = -0.455$, $p < 0.001$), and hypertension remained a strong independent predictor (OR 2.3, 95% CI 1.0–5.0) [52,53]. Early-life disadvantage, including poor childhood health (OR 1.17) and low socioeconomic status (OR 1.39), was associated with later cognitive impairment [44]. Additionally, frailty risk was increased among individuals with cognitive deficits, including global cognitive impairment, verbal fluency impairment, and word list memory impairment [64]. Oral health was also implicated, with greater anterior tooth loss and reduced occlusal support more frequently observed among older adults with cognitive impairment compared with cognitively normal peers [73]. Muhammad et al. (2024) [54] found that active smoking (≥20 pack-years) and passive smoking increased dementia risk (aOR 1.61), while low education (AOR 4.79) and diabetes (AOR 3.23) were particularly strong predictors. Psychosocial factors such as being unmarried, low social participation, and life dissatisfaction significantly predicted mild cognitive impairment [55].

Environmental and social factors also influenced cognitive outcomes. Strong family and peer support were associated with higher quality of life and milder dementia severity [49], whereas social disengagement and rural residence increased risk [40]. Living environment further shaped cognitive outcomes, with community-dwelling older adults demonstrating higher cognitive performance than nursing home residents, and distinct sociodemographic correlates of cognitive decline observed across settings [74]. Exposure to pesticides and nutrient deficiencies in rural and tribal communities were linked to neurodegenerative conditions [69], including Parkinsonism–dementia [56]. Comorbid neurological and medical conditions were common. Among post-stroke patients, 44.7% experienced cognitive impairment [57], and epilepsy and Human Immunodeficiency Virus (HIV) infection were associated with impairment in 83.9% and 47.3% of cases, respectively [58,59]. White matter lesions, poor sleep quality, and hypertension were additional predictors of cognitive decline [60,61].

**Theme 4: Dementia diagnosis.** Sixteen studies [76–91] examined dementia diagnostic tools, with most adopting cross-sectional designs (n = 14) and limited longitudinal validation. A modified Mini-Mental Test demonstrated high sensitivity (0.98) and positive predictive value (0.94) using a cut-off of 33/55, with performance influenced by education level and vascular dementia subtype [76]. A decision-tree algorithm combining neurological assessments with cognitive tests such as the CERAD Battery and Clock Drawing Test effectively distinguished mild cognitive impairment (MCI) from normal cognition, identifying subjective memory complaints and physical exercise history as key predictors [77]. The Dementia Severity Rating Scale showed strong cross-cultural validity [78], while the Mini-Cog achieved high sensitivity (83.2%) but low specificity (49.2%), with completion challenges for non-dementia participants [79]. Olfactory assessment using culturally familiar scents (e.g., coffee, lemongrass) demonstrated feasibility and potential as a low-cost cognitive screening tool in community-dwelling older adults [80].

More resource-intensive diagnostic approaches were less common. Quantitative EEG demonstrated high potential for detecting cognitive impairment in post-stroke patients. One study achieved 96% classification accuracy in identifying

vascular dementia using EEG-based feature extraction [81], while another applied spectral power complexity analysis across working memory EEGs, showing significant differences in spectral entropy between cognitively normal participants, post-stroke patients with MCI, and post-stroke patients with dementia, with entropy values decreasing as dementia severity increased [90]. MRI-based studies reported mixed findings: medial temporal atrophy and Koedam scores were negatively correlated with cognitive function in one study [82], whereas another found no significant associations, likely reflecting limited sample size and statistical power [83]. FDG-PET imaging demonstrated significant correlations between posterior cingulate metabolic activity and MMSE scores, suggesting potential utility for assessing Alzheimer's severity [89]. A multimodal diagnostic model combining olfactory testing, pupillary responses, plasma BDNF levels, and APOE ε4 status improved diagnostic precision for amnestic MCI [84]. Analysis of recall patterns in the Hopkins Verbal Learning Test (HVLT) further enhanced screening sensitivity [85].

Cross-cultural validation studies supported the adaptation of global tools for local use. The Rowland Universal Dementia Assessment Scale (RUDAS) showed good reliability and age sensitivity [86], while the Brief Cognitive Screening Battery (BCSB-INA) demonstrated excellent internal consistency (Cronbach's α = 0.968) and construct validity (KMO = 0.889), confirming its suitability for early dementia detection [87,91]. The Visual Cognitive Assessment Test (VCAT) effectively distinguished healthy controls from cognitively impaired individuals across Southeast Asia without translation or cultural modification [88].

**Theme 5: Dementia treatment, care and support.** Seventeen studies [29,92–107] investigated therapeutic and intervention strategies for dementia care in Indonesia, encompassing five cross-sectional studies [92–96], five quasi-experimental designs [97–101], two randomised controlled trials (RCTs) [102–103], two qualitative studies [29,104], two cohort studies [105–106], and one mixed-methods study [107]. Research remains constrained by small sample sizes, short intervention durations, and limited longitudinal follow-up. Most studies focused on cognitive outcomes, with fewer addressing quality of life, caregiver burden, or implementation feasibility.

Pharmacological and non-pharmacological interventions demonstrated promising cognitive benefits. An RCT showed that citicoline significantly improved MMSE scores among individuals with multi-infarct dementia compared to placebo [102]. Cognitive and physical training interventions were also effective; quasi-experimental and RCT studies reported significant improvements in cognitive performance following structured physical exercise, reading therapy, and combined Brain Gym [97] and slow deep-breathing exercises [103,105,106]. A dietary intervention found that tempeh consumption, particularly variants with lower microorganism content, enhanced language function and overall cognition [105]. Cognitive stimulation therapy (CST) combined with exercise yielded greater improvements in registration and construction domains than CST alone, suggesting the value of multimodal engagement [101]. Physical activity interventions, such as daily jogging, were also associated with increased Brain-Derived Neurotrophic Factor (BDNF) levels and improved cognitive function [106].

Digital and telehealth-based interventions were explored in several feasibility and mixed-methods studies [100,107]. Home-based tele-exercise programmes were reported to be acceptable and beneficial, though participation was influenced by digital literacy, caregiver support, and internet access [100,104]. Mobile health (mHealth) applications designed for PwD and carers showed that perceived usefulness and ease of use positively influenced actual app utilisation, with PwD valuing social connectedness and carers prioritising information on dementia symptoms and treatment [93,107]. Cognitive stimulation therapy delivered via a mobile app received positive feedback from both PwD and carers, demonstrating good feasibility and user acceptance [107]. Online-delivered exercise programmes were similarly acceptable and beneficial, with seven key themes emerging, i.e., motivation, benefits for PwD, impact on carers, challenges and enablers, carer strategies, relationships and support, and receptiveness to online delivery, underscoring the critical role of carer involvement in sustained engagement [104].

Community- and carer-focused interventions were also represented [94,95]. The WHO iSupport programme was successfully adapted for Indonesian carers through translation, expert consultation, and focus groups, showing strong

acceptability and cultural relevance [96]. A community-based dementia awareness intervention using an Asset-Based Community Development (ABCD) approach improved dementia knowledge among participants, with 66% achieving moderate and 28% good knowledge levels following audiovisual training [99]. Studies on strategic planning and cultural adaptation highlighted the importance of contextual tailoring. The STRiDE toolkit was adapted for Indonesia through translation and cognitive interviews, enabling use across diverse community and care settings [92]. Additionally, a Theory of Change approach was employed to co-develop dementia care strategies across seven middle-income countries, including Indonesia, facilitating stakeholder engagement and alignment of national priorities [29].

**Theme 6: Support for dementia carers.** Among 12 studies [15,16,108–117] under the theme support for dementia care, 9 were cross-sectional [16,108–114], three qualitative [15,115,116], and one mixed-methods [117], with only one national [108] and three multisite studies [19,111,112], underscoring the absence of longitudinal and rural-based evidence. Qualitative research highlighted the emotional and relational complexity of caregiving. Studies found that dementia carers were generally younger (mean age ≈ 42 years) than carers for other chronic illnesses and often described caregiving as a journey from denial to acceptance, characterised by emotional strain, perseverance, and evolving understanding of dementia [15,116]. Many carers struggled to obtain a formal diagnosis due to stigma and limited healthcare access, and knowledge gaps significantly shaped caregiving approaches and emotional resilience.

Quantitative studies provided complementary insights into caregiver burden, burnout, and well-being. Approximately 54% of carers reported low emotional fatigue, 33% moderate, and 12% high, suggesting that while most did not experience severe burnout, targeted interventions remain necessary [109–112]. Spousal carers experienced physical, emotional, social, and financial strain but maintained optimism regarding their partner's recovery and family harmony [110]. Unmet needs were widespread, particularly for counselling and psychotherapy (83–92%), and burnout correlated strongly with perceived burden, while limited dementia information had less direct impact [111]. Carers identified needs spanning psychological support, healthcare access, and spiritual or cultural guidance, emphasising the importance of multi-professional collaboration in dementia care.

Studies on knowledge and educational gaps indicated that carers generally had moderate dementia knowledge, with higher scores in care considerations but persistent weaknesses in communication and behavioural management [112]. Knowledge was positively associated with younger age, child–parent caregiving relationships, and previous dementia education, reinforcing the need for structured, targeted educational interventions. Research also linked caregiving to quality of life (QoL) outcomes. Carer burden, personal health status, and family history of Alzheimer's were significant predictors of QoL, while those providing >16 hours of daily care reported markedly poorer well-being.88,89 Social support mitigated depressive symptoms and mediated the relationship between burden and QoL, highlighting its buffering role [111,112]. The severity of cognitive impairment among people with dementia did not significantly affect caregiver stress, burden, or knowledge, with most carers experiencing mild depression and mild-to-moderate burden but maintaining adequate dementia knowledge [116].

**Theme 7: Dementia research and innovation.** Complementing epidemiological and clinical studies, emerging research on biomarkers and pathophysiology has expanded in recent years, reflecting a growing effort to elucidate the biological mechanisms and early indicators of dementia within the Indonesian context. It includes 11 studies [118–128]: seven cross-sectional [118–124], two case reports [125,126], one case-control [127] and one cohort study [128]. These studies were conducted in diverse locations, including Jakarta (N = 4) [118,119,121,123], Bali (N = 2) [124,128], Papua (N = 1) [126], Banten (Java) (N = 1) [122], West Sumatra (Padang) (N = 1) [127], a multisite study within Indonesia (N = 1) [120], and one multinational study [125].

Genetic and amyloid biomarkers were the most frequently explored. Studies of the APOE ε4 genotype found that carriers typically had normal MMSE scores (≥25) but lower verbal memory performance, indicating selective cognitive vulnerability [123]. Further research linked APOE ε4 to a higher likelihood of Alzheimer's disease and elevated plasma Aβ42 concentrations [122], whereas others observed lower plasma Aβ1–42 levels among individuals with amnestic MCI,

yielding moderate diagnostic accuracy (sensitivity 64.8%, specificity 71.9%) but no significant association with APOE ε4 status [119]. These findings suggest that peripheral amyloid markers may aid early detection, though results remain inconsistent across small cohorts.

Oxidative stress and inflammatory pathways have also been examined. Patients with atrial fibrillation and cognitive impairment exhibited reduced Aβ42 and elevated malondialdehyde (MDA) levels, underscoring oxidative stress as a potential contributor to neurodegeneration [127]. In a cohort study of mild traumatic brain injury (MTBI), elevated interleukin-1β (IL-1β) levels were associated with more than twofold increased risk of cognitive decline, supporting the role of inflammatory markers in dementia pathogenesis [128]. Metabolic and hormonal factors were also implicated. In geriatric samples, high LDL-C, cystatin C, and HbA1 levels were associated with moderate-to-severe cognitive impairment, suggesting a link between metabolic dysregulation and cognitive decline [120]. Among postmenopausal women, elevated follicle-stimulating hormone (FSH) levels correlated with mild cognitive impairment, indicating hormonal transitions as a potential biomarker for early detection [118].

Neurophysiological and imaging-based approaches further broadened the research landscape. Pupillary response measures, including pupil size and constriction velocity, were identified as objective, operator-independent biomarkers for early Alzheimer's detection [121,124]. MRI-based studies found that the midbrain-to-pons ratio and the width of the middle cerebellar peduncle were significant predictors of cognitive impairment, correlating positively with MoCA-Ina scores, whereas other brainstem structures showed no significant association [124]. At the frontier of rare and region-specific neurodegenerative conditions, case studies reported atypical presentations of amyotrophic lateral sclerosis–Parkinsonism–dementia (ALS/PDC) in Papua and the Kii Peninsula, linking these to possible environmental–genetic interactions such as soil mineral deficiencies and familial susceptibility [126]. Comparative analyses further revealed that only 5.5% of Asian patients with frontotemporal lobar degeneration (FTLD) reported a family history, compared with 20% in Western populations, suggesting genetic and environmental heterogeneity in disease patterns [125].

## Discussion

This scoping review summarises existing evidence on Alzheimer's disease and dementia in Indonesia, aligning with the WHO Global Action Plan on the Public Health Response to Dementia (2017–2025) [21], and highlights patterns in study design, geographic distribution, and thematic focus across the Plan's seven strategic domains. The temporal distribution of studies highlights that dementia research in Indonesia only began to expand substantially after 2016, with only 6.7% published prior to that year, coinciding with the introduction of Indonesia's National Dementia Plan and the WHO Global Action Plan on Dementia. Despite this growth, post-2016 research remains heavily concentrated in cross-sectional risk factor studies, with comparatively limited evidence evaluating policy implementation, health system integration, or population-level impact. Furthermore, it is disproportionately concentrated in urban areas, particularly on the island of Java. This geographical bias limits the generalisability of findings and obscures the true burden of dementia in rural and eastern regions, where access to healthcare and diagnostic services may be limited. The current review identified that a large proportion of Indonesian studies on dementia were cross-sectional (64.8%) and used non-probability sampling methods (71.6%), limiting the potential to ascertain causal relationships. These results are consistent with reviews of other countries [129,130], indicating the need for more substantial research designs to build a better understanding of dementia and its implications across populations. Intervention research, whether pharmacological or non-pharmacological, is particularly limited, impeding the development of contextually appropriate, evidence-based care.

Risk factor research in Indonesia has focused on global indicators such as lower socioeconomic status, smoking, hypertension, and diabetes. However, context-specific determinants, such as micronutrient deficiencies, malnutrition, neurotoxic exposures, and cultural stigma, are underexplored. Many identified associations stem from cross-sectional studies, limiting clarity on whether these factors are causal, symptomatic, or confounded by preclinical dementia changes. Moreover, advances in diagnostic tools, such as the adaptation of the Montreal Cognitive Assessment (MoCA-INA),

neuroimaging, and biomarker studies, have emerged in recent years. Yet, their use in routine clinical settings is rare, primarily due to cost, lack of validation in local populations, and workforce shortages. These limitations are especially acute in rural and underserved areas, where misdiagnosis or under-recognition of dementia is likely. Compounding these issues, Indonesia, like many LMICs, faces a severe shortage of neurologists, geriatricians, and trained primary care professionals equipped to manage dementia [131,132].

Caregiving is an overlooked but critical component of dementia care in Indonesia. Women carry a disproportionate share of caregiving duties and experience heightened emotional, financial, and physical consequences [133,134]. While qualitative evidence has captured caregiver distress, few studies have evaluated formal caregiver training, support programs, or scalable models of respite care. Digital and community-based interventions (e.g., exercise programs, cognitive stimulation) show promise but require more rigorous evaluation, particularly in low-resource settings where barriers like internet access and digital literacy persist.

Despite the launch of Indonesia's National Dementia Plan in 2016 [11], its implementation has been inconsistent. There is little evidence of integration into primary care, limited community outreach, and no systematic approach to early detection or care coordination. The absence of policy monitoring or evaluation mechanisms hinders assessment of its impact. Indonesia must address key methodological and contextual challenges identified through this review to strengthen dementia research and care. This review identified key methodological and contextual challenges that limit the scope of dementia research in Indonesia. Building on these findings, we propose a set of priorities and recommendations to guide future research and practice (**Table 3**). These are not results from individual studies but rather a synthesis of lessons learned across the literature.

Limitations of the scoping review approach should be acknowledged. Unlike systematic reviews, scoping reviews typically do not assess the quality or risk of bias of included studies, limiting the ability to judge the strength of the evidence. Furthermore, the broad nature of scoping reviews may overlook subtle nuances in study findings. While this review aimed to be comprehensive, some relevant studies may have been missed due to language restrictions or publication bias. Finally, The small number of studies published before 2016 limits meaningful comparison across long historical periods; consequently, temporal interpretations focus on descriptive trends rather than causal change.

## Conclusion

This scoping review integrates the existing state of Alzheimer's and dementia research in Indonesia, identifying an expanding corpus of research and gaps in geographic coverage and methodological soundness. Existing evidence reaffirms traditional dementia risk factors (e.g., age, low education, hypertension, diabetes) and suggests context-specific determinants, such as nutritional deficiencies, environmental exposures, and culturally mediated caregiving responsibilities. Though cross-sectional design dominance and studies concentrated in Java limit broader generalisability and causality, main challenges persist across the continuum of care. While cognitive screening instruments and biomarker-based diagnostic tests are promising, their application in clinical practice is limited by cost, unproven validity in local populations, and weak health system infrastructure. The intensive informal caregiving strains reflect the lack of structured support systems, training, and respite care. Such issues highlight the need for integrated community-based models of dementia care tailored to Indonesia's social and economic context. While the 2016 National Dementia Plan was initiated, policy translation remains fragmented, with minimal integration into primary care and limited outcomes tracking. The response to dementia will be enhanced by investing in longitudinal research, extending studies beyond urban centres, and creating scalable, culturally grounded interventions.

Policy formulation must be guided by empirical assessment and underpinned by multisectoral partnerships among government, academic, and healthcare institutions. Progress in dementia research and care in Indonesia will rely on scientific advances and concerted efforts to promote equitable access, community involvement, and sustainable system-level reforms.

**Table 3. Proposed research priorities and recommendations.**

| Domain | Key Issues & Challenges | Research Priorities & Recommendations |
|---|---|---|
| Data & Research Infrastructure | • Poor record-keeping and data management undermine study integrity.<br>• Over-reliance on cross-sectional studies. | Strengthen data systems to support longitudinal and cohort studies, particularly in underrepresented rural and eastern regions. |
| Assessment Tools | • Limited availability of culturally and linguistically validated tools.<br>• Inconsistent use of appropriate cognitive measures in clinical trials. | • Validate and adapt tools like MoCA-INA and MMSE across diverse populations and dialects.<br>• Develop unbiased culturally adapted low-cost screening tools for use in primary care. |
| Intervention Development | • Barriers to participation due to mobility, low digital literacy, and chronic comorbidities.<br>• Digital literacy and connectivity issues hinder the implementation of non-pharmacological interventions. | • Develop inclusive recruitment strategies targeting older adults and those with chronic conditions (e.g., T2DM).<br>• Evaluate both pharmacological and non-pharmacological interventions for feasibility, scalability, and technological adaptability (e.g., SMS-based support). |
| Health Promotion & Prevention | • Lack of dementia awareness and knowledge in the general population.<br>• Lack of dementia training and low level of confidence in the healthcare professional. | • Design and evaluate community-based health education programs focused on behaviour change, early diagnosis, and caregiver outcomes, especially in low-resource settings.<br>• Evaluate the effectiveness of dementia training programs across different healthcare levels.<br>• Develop scalable, culturally tailored training modules to improve knowledge, confidence, and care quality, particularly for primary care providers. |
| Policy Integration & Systems Research | • Weak integration of research into policy and practice. | • Evaluate the implementation of the 2016 National Dementia Plan, including integration into primary care and UHC.<br>• Strengthen collaboration with policymakers, healthcare providers, and community organisations. |
| Caregiver & Community Support | • Current support networks are informal and inconsistent.<br>• Carers' QoL affected by health status, caregiving burden, and patient factors (e.g., BPSD, gender of care recipient). | • Develop and test culturally resonant, low-cost psychosocial and social support interventions for caregivers, particularly women.<br>• Explore dementia-friendly community models and task-shifting strategies. |
| Biological & Genetic Research | • Investigate APOE ε4, micronutrient deficiencies, and metabolic risks across diverse ethnic populations, ensuring ethical, locally relevant research. | |

## Supporting information

**S1 File. PRISMA Checklist.** PRISMA-ScR (Preferred Reporting Items for Systematic Reviews and Meta-Analyses extension for Scoping Reviews) checklist. Reproduced from Tricco AC et al. (2018) under the Creative Commons Attribution 4.0 International License (CC BY 4.0). See https://www.prisma-statement.org/citing-prisma-2020.
(DOCX)

**S1 Table. Characteristics of included studies, including author, year, study category, dementia type, study type, location, sample size, and settings.**
(DOCX)

## Author contributions

**Conceptualization:** Anna Tjin, Fasihah Irfani Fitri, Sarah Bauermeister, Iracema Leroi, Asri Maharani.

**Data curation:** Anna Tjin, Michael Maitimoe, Diany Syafitri, Shofia Mawaddah, Tung Le, Asri Maharani.

**Formal analysis:** Fasihah Irfani Fitri, Michael Maitimoe, Diany Syafitri, Shofia Mawaddah, Tung Le, Asri Maharani.

**Funding acquisition:** Sarah Bauermeister.

**Investigation:** Fasihah Irfani Fitri, Robert Stewart, Asri Maharani.

**Methodology:** Sarah Bauermeister, Robert Stewart, Asri Maharani.

**Project administration:** Anna Tjin.

**Resources:** Firda Aminy Ma'ruf, Iracema Leroi.

**Supervision:** Sarah Bauermeister, Robert Stewart, Iracema Leroi, Asri Maharani.

**Visualization:** Tung Le, Firda Aminy Ma'ruf.

**Writing – original draft:** Anna Tjin, Iracema Leroi, Asri Maharani.

**Writing – review & editing:** Anna Tjin, Firda Aminy Ma'ruf, Asri Maharani.

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
