## [Editor Report · Decision Letter 0]

13 Jan 2026

PGPH-D-25-03260

Mapping Dementia Research in Indonesia: A Scoping Review of Evidence, Gaps, and Future Directions

Dear Dr. Tjin,

Thank you for submitting your manuscript to PLOS Global Public Health. After careful consideration, we feel that it has merit but does not fully meet PLOS Global Public Health’s publication criteria as it currently stands. Therefore, we invite you to submit a revised version of the manuscript that addresses the points raised during the review process.

We look forward to receiving your revised manuscript.

Kind regards,

Damen Haile Mariam, MD, MPH, PhD

Academic Editor

Journal Requirements:

1. Please upload separate figure files in .tif or .eps format. Also, remove the figures from your manuscript file but keep the legends.

2. We have noticed that you have uploaded Supporting Information files, but you have not included a list of legends. Please add a full list of legends for your Supporting Information files after the references list.

4. Some material included in your submission may be copyrighted. According to PLOS’s copyright policy, authors who use figures or other material (e.g., graphics, clipart, maps) from another author or copyright holder must demonstrate or obtain permission to publish this material under the Creative Commons Attribution 4.0 International (CC BY 4.0) License used by PLOS journals. Please closely review the details of PLOS’s copyright requirements here: PLOS Licenses and Copyright. If you need to request permissions from a copyright holder, you may use PLOS's Copyright Content Permission form.

Potential Copyright Issues:

a. Figure 2: please (a) provide a direct link to the base layer of the map (i.e., the country or region border shape) and ensure this is also included in the figure legend; and (b) provide a link to the terms of use / license information for the base layer image or shapefile. We cannot publish proprietary or copyrighted maps (e.g. Google Maps, Mapquest) and the terms of use for your map base layer must be compatible with our CC-BY 4.0 license.

Additional Editor Comments (if provided):

General Comments:

- The manuscript seems relevant ant timely given the period for “WHO Global Action Plan on Dementia (2017–2025)” is just being completed.

Methods -

- Given the authors' intention to evaluate the existing evidence in line with “WHO Global Action Plan on Dementia (2017–2025)”, the search should probably focus on the period 2015-2025; or should the authors be interested to compare the evidence before and after WHO Global Action Plan, the interval between 2010-2015 and 2016 to 2015 would be more appropriate and informative. Doing so would enable the authors to objectively evaluate the achievements made so far against the “National strategy management of Alzheimer’s and other dementia diseases: towards healthy and productive older persons” and WHO Global action plan 2017 -2025. In addition, the depth of information that can be potentially gained from earlier studies appears to be minimal given there were only a handful of publications for the period 1982-2010.

Results -

- There are major shortcomings in the following two areas.

- 1. Missing/unaccounted references: references 62 -75; and 89-91 were not cited anywhere in the article.

- 2. There are citations that are contradictory such as:

- The title of reference number “44’” does not match with frailty while reference “64” may be more fitting for this citation.

- Reference number 80 has been cited for “innovative olfactory screening and FGD-PET imaging correlation.”

- There also appears to be fragmentation in putting together studies by their category of references which makes difficult to conceptualize and understand the information presented under themes 4 and 5.
---

## [Editor Report · Decision Letter 1]

16 Feb 2026

Mapping Dementia Research in Indonesia: A Scoping Review of Evidence, Gaps, and Future Directions

PGPH-D-25-03260R1

Dear Ms Tjin,

We are pleased to inform you that your manuscript 'Mapping Dementia Research in Indonesia: A Scoping Review of Evidence, Gaps, and Future Directions' has been provisionally accepted for publication in PLOS Global Public Health.

Best regards,

Damen Haile Mariam, MD, MPH, PhD

Academic Editor